# Solution-processable polymers of intrinsic microporosity for gas-phase carbon dioxide photoreduction

Floriana Moruzzi[1], Weimin Zhang[2], Balaji Purushothaman[2], Soranyel Gonzalez-Carrero[3], Catherine M. Aitchison[1], Benjamin Willner[1], Fabien Ceugniet[1], Yuanbao Lin[1], Jan Kosco[2], Hu Chen[4], Junfu Tian[1], Maryam Alsufyani[1], Joshua S. Gibson [5], Ed Rattner[6], Yasmine Baghdadi[6], Salvador Eslava [6], Marios Neophytou[2], James R. Durrant [3], Ludmilla Steier [1] & Iain McCulloch [1] ✉

Four solution-processable, linear conjugated polymers of intrinsic porosity are synthesised and tested for gas phase carbon dioxide photoreduction. The polymers' photoreduction efficiency is investigated as a function of their porosity, optical properties, energy levels and photoluminescence. All polymers successfully form carbon monoxide as the main product, without the addition of metal co-catalysts. The best performing single component polymer yields a rate of 66 µmol h$^{-1}$ m$^{-2}$, which we attribute to the polymer exhibiting macroporosity and the longest exciton lifetimes. The addition of copper iodide, as a source of a copper co-catalyst in the polymers shows an increase in rate, with the best performing polymer achieving a rate of 175 µmol h$^{-1}$ m$^{-2}$. The polymers are active for over 100 h under operating conditions. This work shows the potential of processable polymers of intrinsic porosity for use in the gas phase photoreduction of carbon dioxide towards solar fuels.

Photocatalytic conversion of carbon dioxide into solar fuels is a promising strategy to decrease dependency on fossil fuels, and a promising way to utilise captured carbon dioxide from fossil-fuel burning power plants[1]. Photocatalysis provides reaction pathways to drive thermodynamically unfavourable reactions by using light[2,3]. Currently, carbon dioxide is widely used in methane synthesis in the Sabatier process[4] and in reverse water-gas shift to produce carbon monoxide, which is often further converted into other hydrocarbons such as methanol in the Fischer-Tropsch process. These currently require high temperatures and pressures[5]. Photocatalytic reduction of carbon dioxide offers the opportunity to carry out these reactions at room temperature and minor overpressure. This also gives the

opportunity to shift the reaction equilibrium towards products in this normally exothermic reaction. This work focuses primarily on the formation of carbon monoxide, which is the main intermediate for most pathways to solar fuels, as well as being essential in the production of hydrocarbons and syngas (CO:H$_2$) which can be used in the hydroformylation of alkenes and as turbine fuel[6]. Using CO$_2$ and H$_2$ together to form fuels is already a commercial process, but more sustainable ways of using these reactions are gaining momentum[7].

Inorganic semiconductors are currently at the forefront of performance in carbon dioxide photocatalysis[8–11]. The highest performing is gas-phase photocatalysis are those exhibiting high porosity, a

[1]Department of Chemistry, Oxford University, Chemistry Research Laboratory, 12 Mansfield Road, Oxford OX1 3TA, UK. [2]KAUST Solar Centre, King Abdullah University of Science and Technology (KAUST), 23955 Thuwal, Kingdom of Saudi Arabia. [3]Department of Chemistry and Centre for Processable Electronics, Imperial College London, 80 Wood Lane, London W12 7TA, UK. [4]School of Physical Sciences, Great Bay University, Dongguan 523000, China. [5]Henry Royce Institute Oxford Centre for Energy Materials Research, Department of Materials, University of Oxford, Parks Road, Oxford OX1 3PH, UK. [6]Department of Chemical Engineering, Imperial College London, London SW7 2AZ, UK. ✉e-mail: iain.mcculloch@chem.ox.ac.uk

property which has been shown to enhance properties of $CO_2$ adsorption and activation, increased charge separation and charge transfer, increased gas diffusion, and reaction rate[12]. Despite this, inorganic semiconductors are limited by their poor energetic tunability, important for the optimisation of light harvesting and conduction and valence band placement to optimise photocatalytic activity, and difficulties in processing. In liquid phase photocatalysis, organic materials have been investigated, such as COFs[13–18], hyper-crosslinked polymers[19], and carbon nitrides[20–22]. The synthetic preparation of these materials, however, typically results in an intractable product, limiting their processability. Molecular photocatalysts have also been tested but have suffered from difficulty in preparation and scalability, low processability and low stability[23]. Comparatively little attention has been given to solution processable semiconducting polymers for this application, though their solution processability, allowing for tunability of energy levels, warrants an increased interest in this area[24]. Porous organic materials have shown promise for applications in gas separation and storage, and the possibilities they might offer in photocatalysis, especially in the gas phase[25]. Using hydrogen as a hole scavenger in the gas-phase reactions bypasses the need for liquid phase scavengers such as triethanolamine (TEOA) and ascorbic acid, it also addresses the issues of poor solubility of carbon dioxide in aqueous electrolyte solutions. Furthermore, thermodynamically the reduction potential to generate hydrogen from water is lower than the reduction potential of $CO_2$, so hydrogen evolution is normally favoured in water assisted $CO_2$ photoreduction. However, using hydrogen directly as a scavenger maximises the $CO_2$ reduction process[26]. Gas-phase photoreduction also has the potential to offer increased active area and more rapid diffusion of reagents and intermediates[27–29].

Triphenylamine based conjugated polymers have been shown to catalyse $CO_2$ to CO reduction using water vapour, at a rate of 37.15 µmol h$^{-1}$ g$^{-1}$, with no added co-catalyst, and a quantum efficiency of 0.19% at 420 nm[30]. Similarly, naphthalene diimide conjugated polymers have demonstrated $CO_2$ photocatalysis in the gas phase, obtaining CO as the sole product[31]. However, in these cases stoichiometric oxygen or $H_2O_2$ were not detected and so the exact oxidation reaction occurring is unclear. Whilst there are some reports of $CO_2$ reduction being achieved in tandem with oxygen production[32], the majority of organic examples rely on oxidation of a hole scavenger as the accompanying half reaction. Solution processability is important to make full use of the advantages organic materials can offer. Polymers of intrinsic microporosity (PIMs) can offer a combination of solution processability, porosity and conjugation, essential for synthetic design of materials for gas-phase carbon dioxide photoreduction. PIMs comprise a network of interconnected intermolecular voids due to inefficient packing[33]. Due to their microporous nature, PIMs have drawn attention in applications such as gas separation and storage[34], fuel cells[35], use in membranes[36] and energy storage[37]. Linear conjugated polymers based on spiro structures have been recently shown to be successful in hydrogen evolution, these were not soluble but exhibited high performance, with porosity and hydrophilicity being credited as increasing water penetration and therefore hydrogen evolution rate[38,39].

In this work, we developed a series of polymers based on a bulky monomer comprised of a rigid iptycene group. The iptycene group is a three-dimensional, shape-persistent moiety consisting of three aromatic "blades" protruding from a single hinge, which contains a high internal free volume, as shown in Fig. 1a. This sterically hindered group suppresses close intermolecular pi–pi stacking, preventing polymer chains from packing efficiently and generates the "voids" and intrinsic free volume associated with PIMs. The *tert*-butyl group was incorporated in the iptycene units to enhance polymer solubility, necessary to grow the polymer chain during synthesis, as well as to facilitate thin film processing from solution. The bulkiness of the *tert*-butyl group

further inhibits packing, which subsequently increased the overall porosity of the polymers[40]. This monomer was used to synthesise and compare a series of porous polymers in order to take a step towards intelligent synthetic material design for gas-phase photocatalysis. The conjugated polymer backbone gave rise to the optical and electronic properties due to delocalised aromatic pi-electrons along the chain, which can be modified by the electron rich or deficient nature of the co-monomers chosen. The exceptionally high internal surface area arising from the intrinsic porosity of these triptycene based polymers can facilitate an increased number of reaction sites, and the small pore sizes are particularly accessible by gasses. It was therefore promising to use polymers as thin films for photocatalysis in the gaseous state[25,41].

## Results
### Polymer synthesis and characterisation
Four polymers were synthesised in this study, as shown in Fig. 1a, namely pTA, pTA-Ph, pTA-Th and pTA-BT. pTA was synthesised from C-H activation with an iron catalyst from the bulky monomer, whereas pTA-BT and pTA-Ph were synthesised by Suzuki polymerisation with a palladium catalyst, and pTA-Th via Stille, also with a palladium catalyst. The molecular weights obtained from the polymerisations ranged from between 20 and 39 kDa for the co-polymers and 146 kDa for the homopolymer, pTA (see Experimental).

The polymer backbone conformations were estimated via DFT structure optimisations (Table 1). pTA has a dihedral angle of 24.4°, pTA-Th has a more variable bond angle (Table S1), with an average of 37.9°, pTA-Ph and pTA-BT had a dihedral angle of 48.5° and 44.3° respectively. These steric properties were reflected in the materials frontier molecular orbital (FMO) energies; highest occupied molecular orbital (HOMO) energies were measured by photoelectron spectroscopy in air (PESA) (Fig. S10) and the values obtained are summarised in Fig. 1c and Table 1. The HOMO of pTA was measured to be 5.6 eV, which is considerably deep for a thiophene polymer, attributed to the significant dihedral twist between linked thiophene repeat units imposed by the iptycene repeat unit. The addition of a thiophene co-monomer (pTA-Th) led to a shallower HOMO of 5.4 eV, arising from an increase in backbone planarity, since this unsubstituted 5 member ring co-monomer can adopt a more coplanar configuration, eliminating the strong steric twisting between two neighbouring iptycene rings, thus increasing conjugation[42]. pTA-BT had a deeper HOMO energy level than pTA of 5.9 eV, attributed to the electron-deficient nature of the BT co-monomer unit which also strongly stabilised the hybridised LUMO energy level, thus lowering the bandgap. pTA-Ph also exhibited a deeper HOMO energy level than pTA, of 5.7 eV, attributed to the contribution of the lower electron density phenyl co-repeat unit, and the larger dihedral twisting from the four alpha hydrogens on the phenyl ring. This also resulted in a blue-shifted absorption onset of 467 nm[43].

Figure 1b shows the UV–Vis spectra of the polymers, with data summarised in Table 1. The band gap was calculated from the absorption onset in order to obtain the lowest electronic transition accessible via a single photon[44]. Electron affinity (EA) data were calculated from the addition of the optical band gap to the HOMO values obtained from PESA[37]. Figure 1e illustrates that the energy levels of all the polymers are favourable to drive both carbon dioxide reduction to carbon monoxide, and hydrogen oxidation to H$^+$, they were therefore deemed suitable for $CO_2$ photocatalysis testing[45].

The excited state of the porous polymer films was investigated using steady state and time-resolved photoluminescence (PL) (Fig. 1). All polymer films showed broad PL emission bands, with maximum at 467, 550, 563 and 647 nm for pTA-Ph, pTA, pTA-Th and pTA-BT, respectively. These values were red-shifted compared with those observed in solution of polymers in chloroform (Fig. S12). The comparison of the PL intensity on film, showed that the polymer pTA-Ph emission was significantly higher than all other polymers

(Fig. 1c). The analysis of the PL decays in polymer films revealed a multiexponential decay, which is dependent on the emission wavelength (Fig. 1d). The average lifetimes ($t_{avg}$) of the polymers are in the range of 0.2 ns to 0.5 ns measured at the PL emission maximum and up to 2 ns for pTA-Ph measured at red-shifted emission wavelengths as shown in Fig. 1c (and Table S2). In contrast, polymer solutions showed a mono-exponential PL decay (Fig. S12), independent of the emission wavelength, indicating the decay of single emissive species in solution. The reduction of emission efficiency and the increase in the PL lifetime of films and concentrated solutions of the polymers is attributed to the formation of broad density of electronic states due to differences in polymer aggregation[46–49]. Residual catalyst arising from the noble metal-mediated carbon coupling polymerisation reactions has been shown to form metal particles within the bulk of the synthesised conjugated polymer, specifically with palladium[50]. These impurities can detrimentally effect photocatalytic reactions, since they could preferentially facilitate competing side reactions, and in high loadings can parasitically absorb light, or can act as co-catalysts[51]. Scavenging with sodium diethyldithio-carbamate during purification, was carried to minimise the amount of palladium in the polymers[52]. ICP-MS was used to evaluate the amount of residual metal after washing, and the results are summarised in Table 1. Unlike previous studies[50], no correlation was observed between palladium concentrations and rate of carbon monoxide formation (Fig. S13)[53].

Surface area measurements were carried out on the polymer powders. The observed surface areas were between 480 and 860 $m^2 g^{-1}$ which are among the highest achieved for soluble conjugated polymers[54]. The obtained isotherms are shown in Fig 2. and summarised in Table 2. Commonly, linear conjugated polymers have surface areas between 5 and 20 $m^2 g^{-1}$[55]. In all cases, the iptycene containing polymers exhibit at least an order of magnitude higher porosity than an analogous, non-triptyl thiophene polymer, pDHT, synthesised as a non-porous comparison (SI). porosity. The synthesis and properties of this polymer are shown in the SI (Figs. S1–3). pDHT exhibited a porosity of 29 $m^2 g^{-1}$, which is an order of magnitude lower than all porous polymers, and its rate of CO production was significantly lower than that of the porous polymers.

The polymer with the highest BET value was pTA, likely as it had the highest triptyl substitution, with an apparent BET surface area of 862 $m^2 g^{-1}$ and an average pore size of 4 nm. It was followed by pTA-Th,

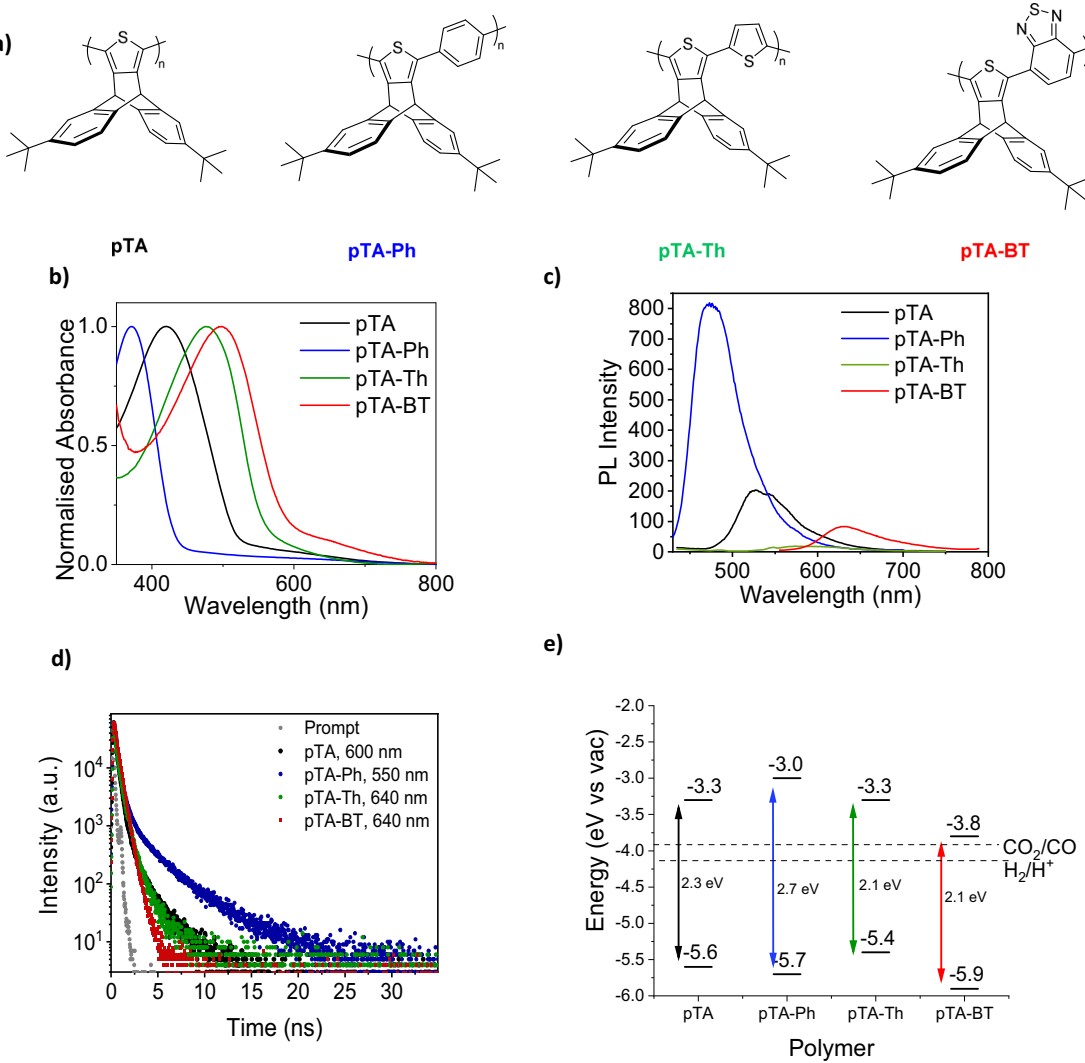

**Fig. 1 | Polymer structure and optical properties. a** Scheme showing the molecular structures of the polymers presented in this study. **b** Normalised UV–Visible thin-film spectra of the polymers. **c** PL spectra of polymers films upon 400 nm excitation, corrected by the number of absorbed photons. **d** Comparison of the PL decay kinetics of polymers films, excited at 404 nm, and recorded at different polymer emission wavelengths, whereby Prompt corresponds to the instrument response function (IRF). **e** Energy level diagram of the four polymers.

with a surface area of 761 m² g⁻¹ and an average pore size of 5 nm. pTA-BT had a surface area of 596 m² g⁻¹ and an average pore size of 4 nm, finally pTA-Ph showed a BET surface area of 487 m² g⁻¹ and an average pore size of 5 nm. The polymers mainly show type II isotherm behaviour which is characterised by a sharp increase at relative pressures below 0.01 which are attributed to micropores, at the knee monolayer formation begins, with multilayer formation along the straight line at medium pressures, and the incline at higher pressure normally representative of larger macropores filling up[56,57]. pTA-BT appears to have the largest fraction of micropores, with an average pore diameter of 4 nm. pTA exhibits a high fraction of micropores, but also showed some macropores with a peak at around 50 nm. pTA-Th showed a wider distribution of mesopores, with a peak around 30 nm. The isotherm of pTA-Ph shows a sharp incline at a relative pressure of 0.9, often indicative of macroporous morphology, further supported by the pore distribution, with an increased volume of pores between 60 and 100 nm.

### Table 1 | Summary of optoelectronic properties of the polymers and DFT bond angles

| Polymer | IP (eV)ᵃ | $E_g$ (eV)ᵇ | EA (eV)ᵃ | PL max (nm) | PL lifetime ($t_{avg}$ᵃ) (ns) | DFT dihedral bond angle (°) |
|---|---|---|---|---|---|---|
| pTA | 5.6 | 2.3 | 3.3 | 550 | 0.75 | 24.4 |
| pTA-Ph | 5.7 | 2.7 | 3.0 | 467 | 2.34 | 48.5 |
| pTA-Th | 5.4 | 2.1 | 3.3 | 563 | 0.56 | 37.9 |
| pTA-BT | 5.9 | 2.1 | 3.8 | 647 | 0.41 | 44.3 |

IP ionisation potential from PESA, EA electron affinity, $E_g$ optical band gap, PL photoluminescence (see S7 for PL lifetimes calculations).
Measurements carried out in:
ᵃEstimated from the EA and $E_g$.
ᵇThin film on ITO spin-coated from 10 mg ml⁻¹ CHCl₃ solution.

## Photocatalysis measurements

Photocatalysis measurements were carried out in a custom set-up. Polymers were tested as films, dropcast from chloroform and subjected to irradiation in an atmosphere of H₂:CO₂ (3:1, 1.15 bar), see SI for full details.

Carbon monoxide and methane were detected by GC. Carbon monoxide was the main product obtained, and therefore the kinetics of CO formation over time were plotted (Fig. 3). CO is the most common first intermediate in the formation of higher hydrocarbons, and was therefore considered a good metric for the efficiency of the materials[58]. Most polymers obtained high CO selectivity (above 98%) with small amounts of methane detected, although pTA produced a higher amount of methane, with 13% and 87% CO selectivity, compared to the others (Fig. S16). The rates obtained were summarised in Table 3, and the kinetic curves shown in Fig. 3. pTA-Ph showed the highest rate, of 66 µmol h⁻¹ m⁻², followed by pTA with 53 µmol h⁻¹ m⁻². pTA-Th and pTA-BT showed slower rates, of 25 µmol h⁻¹ m⁻², and 21 µmol h⁻¹ m⁻², respectively. pDHT was also included in Table 3, showing comparable rate to carbon nitride and was outperformed by all of the porous analogues. The AQY at 450 nm calculated for pTA-Ph was 0.088% (SI), which is lower than CO₂ photocatalysis in the water phase[24], but promising as a single component material. Since different set-ups and testing conditions give vastly different results, polymer performances were calibrated against know and commercially available standards of

### Table 2 | Summarising BET results

| Polymer | BET surface area (m² g⁻¹) | Average pore size (nm) |
|---|---|---|
| pTA | 862 | 4 |
| pTA-Ph | 487 | 5 |
| pTA-Th | 761 | 5 |
| pTA-BT | 596 | 4 |

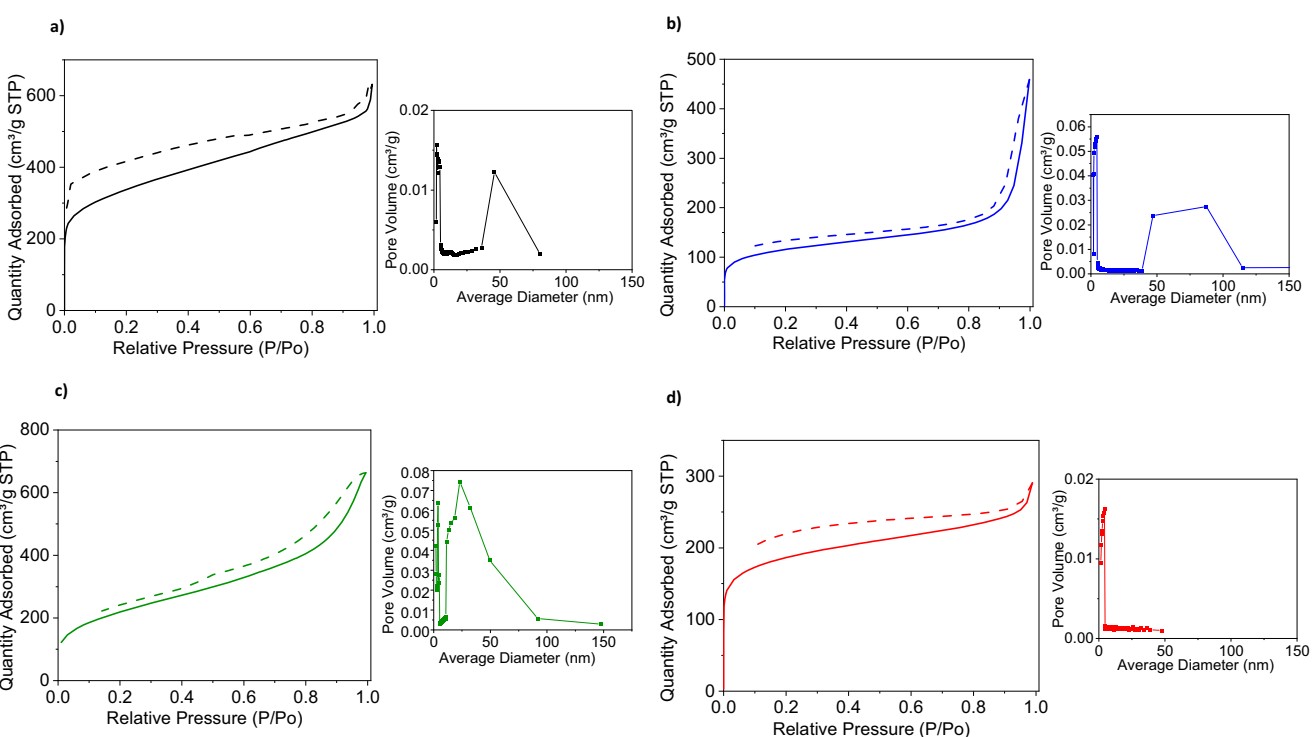

**Fig. 2 | BET isotherms run in nitrogen.** Dotted lines represent desorption, filled lines adsorption, with corresponding average pore diameter graphs. **a** pTA, **b** pTA-Ph, **c** pTA-Th, **d** pTA-BT.

carbon nitrides and titanium dioxide (P25)[59]. Both of these materials are included in Fig. 3. Titanium dioxide ($TiO_2$), dropcast from sonicated $TiO_2$ P25, produced CO at a rate of 21 μmol h$^{-1}$ m$^{-2}$,. All four porous polymers outperformed the titanium dioxide standard. It should be noted that the titanium dioxide standard used in this study shows comparable rates of products to literature results of similar circular photocatalytic reactors in the presence of carbon dioxide and hydrogen[60,61]. Similarly, carbon nitrides, which are organic materials, with a band gap of 2.7 e.V., therefore similar to the polymers in this study energetically, were also outperformed by the four porous polymers[21,62,63]. This supports a good calibration of the testing set-up and evaluated against these standards, the polymers had higher rate of carbon monoxide production.

Photocatalysis can be affected by a multitude of factors, therefore establishing a specific relationship connection between reduction rate and molecular structures is important[64]. Several polymer and film properties were evaluated for their effect on photocatalytic rate and plotted in Fig. 4.

The right-side $y$-axis of Fig. 4 shows the rate of CO production of the polymers. The left side $y$-axis shows the values of parameters observed that could be factors affecting CO production rates. Thus a clear trend in rate could be visualised corresponding to a specific polymer parameter, $t$. The influence of residual palladium content on CO production was also investigated, however the trend of increasing amount of Pd did not match that of increasing CO production rate observed (Fig. 4a and Table S6). Although there was a marked difference between pDHT (the non-porous analogue) and its porous counterparts, the trend of increasing BET surface area did not match the trend of increasing performance for the polymers with intrinsic porosity (Fig. 4b). The distribution of pore sizes did however correlate with CO production, where pTA-Ph, with the highest percentage of larger pores,

i.e. above 50 nm, exhibited the highest reduction rate (Fig. 4c), this could therefore be a contributing factor to reduction rate. It was hypothesised that although smaller pores would still allow for gasses to diffuse, either gas diffusion into active sites or product diffusion out of the active site could be enhanced by interconnected larger pores, larger macropores in inorganic materials have been shown to exhibit increased product formation, and a similar effect could be hypothesised in porous polymers[65]. No trend was observed with the LUMO level of the polymers which governs the overpotential applied within the reaction (Fig. 4d) molecular weight of the polymers (Fig. 4e) also did now show a clear trend. The photoluminescence lifetime decay kinetics (Fig. 4f) was also observed to correlate well with photocatalytic activity, with longer fluorescence lifetimes, correlated to greater photocatalytic performance. In particular, pTA-Ph displayed an almost three-fold increase in photoluminescence lifetime compared to pTA, which could account for its higher photocatalytic activity, despite similar energy levels and a lower surface area. The fact that radiative exciton decay is more dominant in the higher performing polymers is indicative of relatively slower non-radiative decay pathways accessible to these polymers, and slower excited state decay[66]. This is advantageous, because once generated, excited states should undergo photocatalysis more quickly than they recombine in order to successfully utilise incident photons to photoreduce $CO_2$. Indeed, organic semiconductors in OLEDs often utilise similar rigid, sterically hindered motifs in order to increase photoluminescence quantum yield in the solid state[67,68].

In order to verify that the CO measured in the photocatalytic experiments arises from $CO_2$ reduction, and not polymer decomposition, control experiments and isotope labelling experiments were carried out. Control experiments are outlined in the appendix (Fig. S14 and Table S4). The polymers were tested using the following controls (1) Replacing the $CO_2$ with $N_2$ (2) Replacing the $H_2$ with $N_2$ (3) using only $N_2$ (4) with operating conditions gasses but no illumination, 10% or less CO was detected in these control tests, compared to the overall rate obtained using normal operating conditions, supporting that the photoreduction of $CO_2$ has arisen from polymer photocatalysis. To further support this, $^{13}CO$ studies were carried out, as described in the SI and shown in Fig. S17. $^{13}CO$ production was observed after light irradiation overnight with the addition of $^{13}CO_2$. $^{13}CO$ was not observed in a $^{12}CO_2$ run nor in a dark run, further demonstrating the catalytic nature of the polymers.

## Copper based co-catalyst

In order to further optimise the efficiency of the photochemical reduction, the use of a co-catalyst was explored. Copper Iodide (CuI) has previously been reported as a hole injection layer in OPV devices[69], as well as in electrocatalysis[70]. Its transparent nature and high conductivity being promising properties. Addition of 5% CuI (with respect to copper) to polymer solution formulation prior to thin film fabrication led to a higher rate of CO formation, as observed in Fig. 3a. The efficiency trend remained the same, with rates

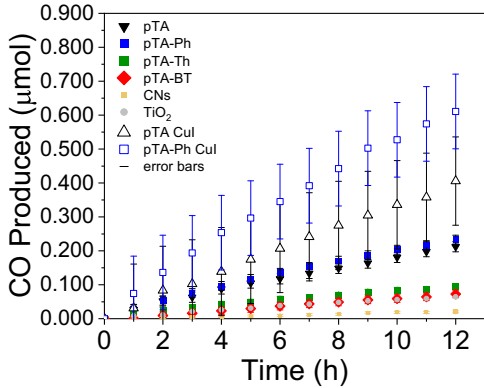

**Fig. 3 | CO production over time from the polymers (2 mg) with no added co-catalyst and with CuI co-catalyst (5% with respect to copper).** $CO_2$ and $H_2$ were in a 1:3 ratio and at 1.15 bar. Standard deviation is shown as error bars in the graph.

**Table 3 | Summary of rates of carbon monoxide production**

| Polymer | Rate (μmol h$^{-1}$) | Selectivity to CO production (%)[a] | Rate (μmol h$^{-1}$ m$^{-2}$) | Rate with added CuI (μmol h$^{-1}$) | Rate with added CuI (μmol h$^{-1}$ m$^{-2}$) |
|---|---|---|---|---|---|
| pTA | 0.017 ± 0.001 | 87 | 53 ± 3 | 0.033 ± 0.003 | 106 ± 10 |
| pTA-Ph | 0.019 ± 0.001 | 98 | 66 ± 4 | 0.05 ± 0.01 | 175 ± 35 |
| pTA-Th | 0.007 ± 0.001 | 98 | 25 ± 3 | 0.016 ± 0.005 | 55 ± 17 |
| pTA-BT | 0.006 ± 0.001 | 99 | 21 ± 4 | 0.015 ± 0.002 | 37 ± 6 |
| pDHT (SI) | 0.002 ± 0.0006 | – | 10 ± 3 | – | – |
| CNs | 0.002 ± 0.0004 | – | 6 ± 2 | 0.002 ± 0.0005 | 6.6 ± 2 |
| $TiO_2$ | 0.005 ± 0.001 | 96 | 21 ± 5 | – | – |

Rates were calculated from the slope of the curves.
[a]Selectivity calculated as a percentage of CO detected vs. $CH_4$.

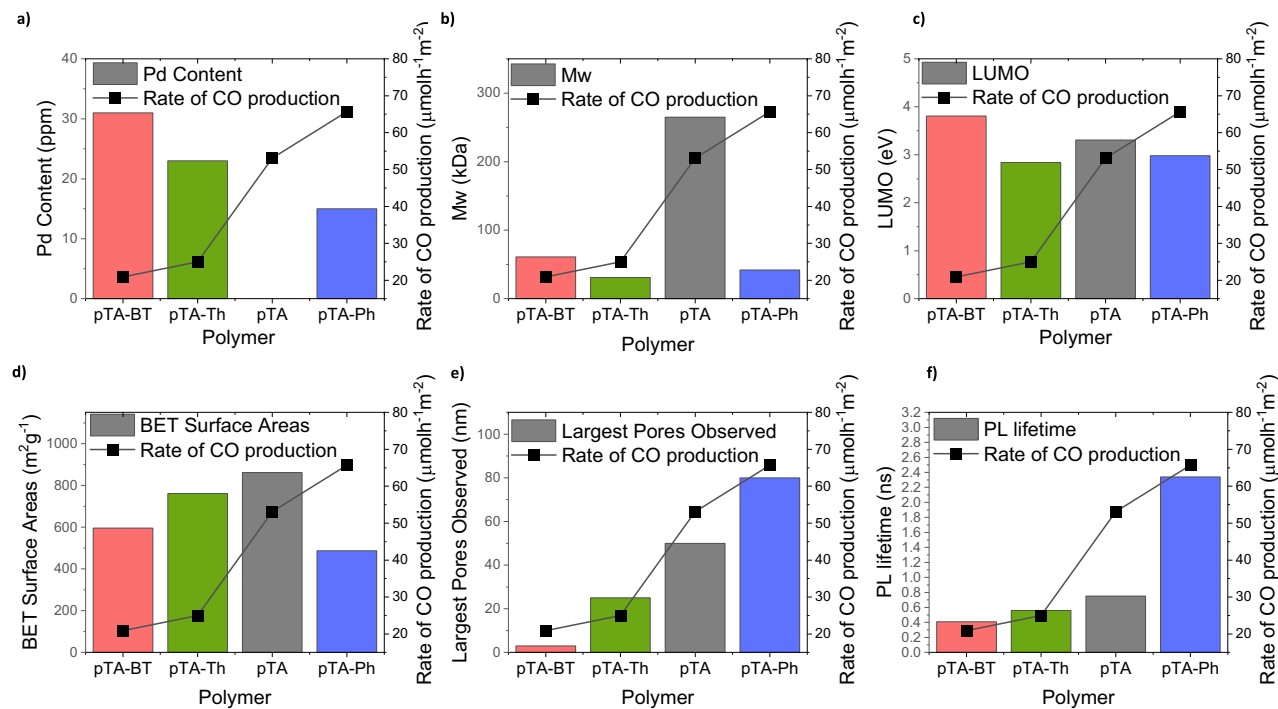

**Fig. 4 | Relation between photocatalytic rate and polymer properties.** Graphs comparing a range of polymer parameters with the rate of CO formation **a** palladium content, **b** BET surface area o, **c** Largest pores observed, **d** LUMO level, **e** molecular weight, **f** Photoluminescence lifetimes.

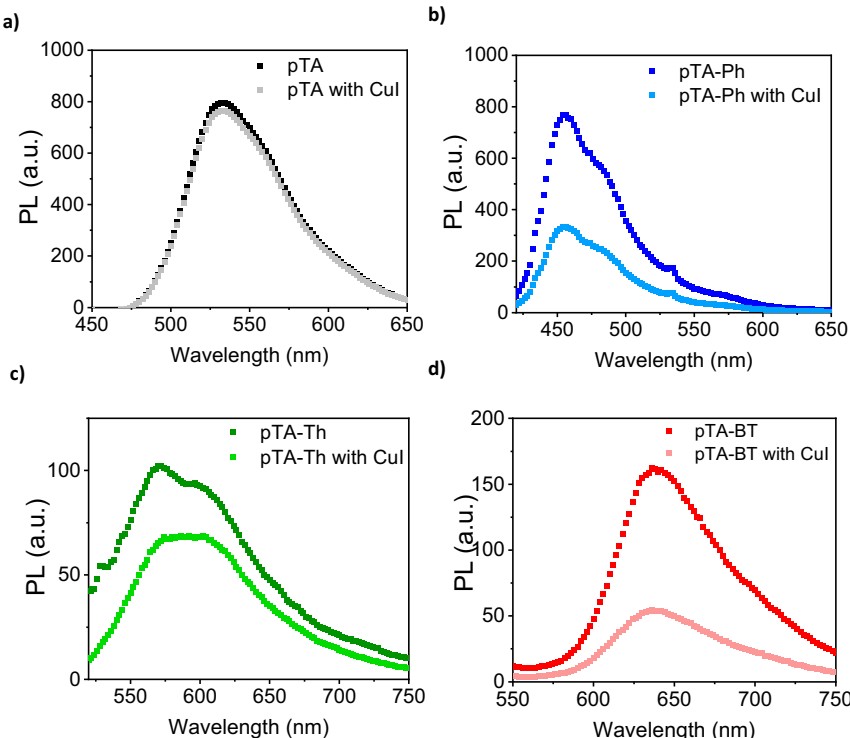

**Fig. 5 | Polymer photoluminescence. a–d** Comparison of the PL spectra of pristine polymers as well as polymers in the presence of CuI, the measurements were carried out in hydrogen to mimic photocatalysis conditions.

increasing by a factor of up to 2.5, with pTA-Ph reaching a rate of 175 µmol h⁻¹ m⁻². Carbon nitride was also tested, but showed minor improvement only. This could be due to CNs being insoluble, and therefore the co-catalyst may not have dispersed as well through the film bulk (Fig. S15).

As shown in Fig. 5a–d, the addition of 5% (w.r.t. Cu) CuI in the drop-casted polymers solution resulted in photoluminescence quenching (PLQ) of the polymers films pTA-BT, pTA-Ph and pTA-Th (PLQ of 35–67%), which suggests there is charge transfer from the polymers to the CuI. The quenching was not observed for pTA in the

a)

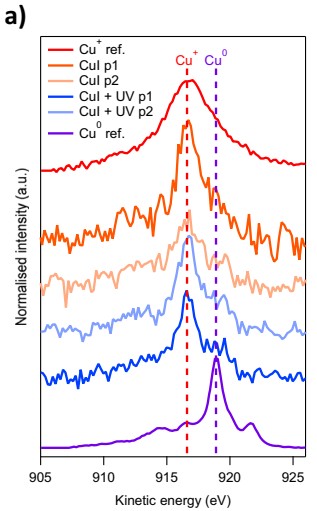

b)

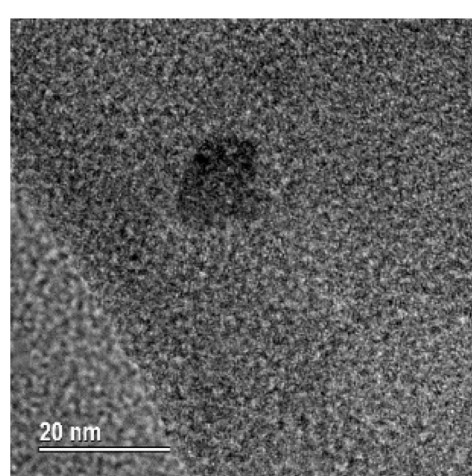

**Fig. 6 | Morphology and valency of copper in polymers. a** XPS data of CuI, compared to CuI treated in the reactor under operating conditions, compared to copper metal reference p1 and p2 are two different points on the surface on which the measurement was focused. **b** TEM image showing copper nanoparticle in pTA-Ph polymer thin film.

sample tested, TEM analysis was carried out on a thin film sample of pTA-Ph after photocatalysis, which exhibited black clusters of around 20 nm, which were attributed to copper nanoparticles, distributed unevenly within the polymer film. These were likely formed during photocatalysis conditions due to the presence of hydrogen in the system facilitating the reduction of Cu(I) to Cu(0). It is likely that these copper metal nanoparticles are acting as the co-catalysts in the reaction, rather than CuI. ICP-MS confirmed the presence of around 5% copper in the polymer films. When CuI was tested by itself, in the same amount added to the polymer samples, no CO was observed, further supporting its role as a co-catalyst.

XPS measurements of copper iodide samples pre- and post-exposure demonstrate differences in the spectral features of the Cu LMM Auger (Fig. 6a) and the Cu 2p photoelectron (Fig. S19) transitions. Following UV + $H_2$ exposure (blue), CuI samples exhibit a 20% increase of a secondary peak at 919.0 eV, relative to the dominant peak at 916.6 eV. This change is consistent with an increase in the presence of Cu(0), as shown by the reference spectrum of argon ion etched copper metal (purple). The same peak is not observed in the samples which were not exposed to operating conditions, supporting the notion that copper metal is being formed during the reaction[71]. Copper bromide (CuBr) was also tested with pTA-Ph, and yielded a rate of 170 µmol h$^{-1}$ m$^{-2}$, this was only slightly lower than CuI added samples (175 µmol h$^{-1}$ m$^{-2}$). The similar increase in the rate of reaction when adding CuI or CuBr to pTA-Ph compared to the pristine polymer suggested that the copper rather than iodine is the important species in the reaction.

## Discussion

Four solution-processable polymers of high intrinsic porosity were synthesised and their efficiency in gas-phase carbon dioxide photoreduction was investigated, in a hydrogen and carbon dioxide atmosphere. It was found that the polymers produced carbon monoxide as the main product with no added metal co-catalyst. pTA-Ph was found to be the highest performing polymer. It was hypothesised that the combination of a higher fraction of macropores combined with a high photoluminescence lifetime and intensity due to inefficient packing in the solid phase, were responsible for its increased reduction rate. Recycling studies confirmed the catalytic behaviour of pTA-Ph. The addition of 5 wt% CuI was found to increase the rate of photoreduction, due to metallic copper formed in the reducing hydrogen atmosphere, acting as a co-catalyst. The polymers were compared against commercially available carbon nitride and titanium dioxide, which were used as benchmarking materials, and the porous polymers each outperformed these. This study takes a step towards synthetic design of organic semiconductors for carbon dioxide photocatalysis, and is one of the first examples to utilise hydrogen oxidation as the co-reaction, which opens us the field of gas-phase $CO_2$ photoreduction, with no need to rely on liquid phase scavengers such as TEOA and ascorbic acid. Overall, it was found that although porosity was important in gas-phase $CO_2$ photoreduction, factors which normally govern solution phase photocatalysis, such as PL life-times also play a vital role.

## Methods

### 2,6-di-*tert*-butyl-9,10-dihydro-9,10-[3,4]furanoanthracene-12,14-dione (2)

A mixture of maleic anhydride (8.1 g, 82.63 mmol) and 2,6-di-*tert*-butylanthracene[72] (20 g, 68.85 mmol) in toluene (190 ml) was heated at reflux overnight. After cooling to room temperature, the precipitate was collected by filtration and washed with cold diethyl ether to give the product as white solid (24 g, 91%).

$^1$H NMR (500 MHz, CDCl$_3$) $\delta$ 7.44 (d, $J$ = 1.9 Hz, 1H), 7.38 (d, $J$ = 1.8 Hz, 1H), 7.34 (d, $J$ = 7.8 Hz, 1H), 7.28 (s, 1H), 7.24 (m, 2H), 4.81 (m, 2H), 3.55 (m, 2H), 1.33 (s, 9H), 1.30 (s, 9H).

$^{13}$C NMR (126 MHz, CDCl$_3$) $\delta$ 170.77, 170.67, 150.89, 150.35, 140.62, 138.04, 137.84, 135.24, 124.66, 124.40, 123.86, 123.72, 122.24, 121.52, 48.32, 48.27, 45.46, 45.44, 34.76, 34.71, 31.47, 31.40.

### (2,6-di-*tert*-butyl-9,10-dihydro-9,10-ethanoanthracene-11,12-diyl)dimethanol (3)

A solution of **1** (22.8 g, 58.7 mmol) in THF (100 ml) was added dropwise via cannula to a stirred suspension of lithium aluminium hydride (5.57 g, 146.7 mmol) in THF (100 ml) at 0 °C. After the addition was complete, the mixture was heated at reflux overnight. After cooling to room temperature, the suspension was diluted with THF (500 ml) and quenched by addition of water (5.6 ml), followed by NaOH (2.2 g in 5.6 ml of water) and finally water (5.6 ml). After hydrolysis was complete, the resulting mixture was filtered, the solids washed with THF and the solvent removed in vacuo to give diol **3** in 98% yield (21.5 g).

$^1$H NMR (400 MHz, CDCl$_3$) $\delta$ 7.33 (d, $J$ = 1.9 Hz, 1H), 7.26–7.20 (m, 2H), 7.17–7.10 (m, 3H), 4.13 (s, 2H), 3.59 (m, 2H), 3.26 (m, 2H), 2.43–2.30 (m, 2H), 1.33 (s, 9H), 1.30 (s, 9H).

$^{13}C$ NMR (101 MHz, CDCl$_3$) $\delta$ 148.83, 148.72, 143.33, 140.70, 140.48, 137.87, 124.07, 122.74, 122.42, 122.38, 121.67, 120.46, 64.70, 64.65, 48.07, 47.98, 43.81, 43.65, 34.60, 34.53, 31.61, 31.59.

### (2,6-di-*tert*-butyl-9,10-dihydro-9,10-ethanoanthracene-11,12-diyl)bis(methylene)bis(4-methylbenzenesulfonate) (4)

Tosyl chloride (12.6 g, 66 mmol) was added slowly in portions to a solution of **3** (10 g, 26.4 mmol) in dry pyridine (50 ml) at 0 °C, with the temperature maintained below 5 °C during the course of addition. The mixture was then warmed to room temperature and stirred overnight. The reaction mixture was then poured onto 2 M HCl (350 ml) and stirred until the precipitate was no longer sticky. The product was collected via filtration and washed with water. Yield: 24 g (60%).

$^1H$ NMR (400 MHz, CDCl$_3$) $\delta$ 7.78 (m, 4H), 7.38 (m, 4H), 7.23 (m, 2H), 7.14 (m, 1H), 7.01 (m, 1H), 6.91 (m, 1H), 4.26 (dd, $J = 16.7$, 1.4 Hz, 2H), 3.74 (m, 2H), 3.38 (m, 2H), 2.51 (s, 3H), 2.48 (s, 3H), 2.42 (m, 2H), 1.3(s, 9H), 1.26(s, 9H).

$^{13}C$ NMR (101 MHz, CDCl$_3$) $\delta$ 149.37, 149.29, 145.05, 142.08, 139.42, 139.36, 136.56, 132.76, 132.66, 130.04, 130.01, 129.94, 127.99, 127.86, 127.83, 124.88, 123.20, 122.93, 122.87, 122.66, 120.91, 69.33, 45.46, 45.40, 40.01, 34.63, 34.51, 31.53, 31.44, 21.70.

### 2,6-di-*tert*-butyl-9,10-dihydro-9,10- [3,4] thiophenoanthracene (5)

To a solution of sodium sulfide nonahydrate (4.9 g, 20.4 mmol) in 40 ml Dimethyl sulfoxide (DMSO) was added dropwise while stirring at 100 °C, a solution of **4** (7 g, 10.19 mmol) in 30 ml of DMSO. The mixture was stirred overnight at 100 °C. The reaction solution was added to crushed ice and the precipitated product was filtered and washed with water to give the product as a white solid (3.45 g, 90%).

$^1H$ NMR (500 MHz, CDCl$_3$) $\delta$ 7.33–7.28 (m, 2H), 7.24–7.18 (m, 2H), 7.17–7.11 (m, 2H), 4.07 (m, 2H), 2.96–2.89 (m, 2H), 2.86–2.78 (m, 2H), 2.27–2.16 (m, 2H), 1.33 (s, 18H).

$^{13}C$ NMR (126 MHz, CDCl$_3$) $\delta$ 148.96, 148.71, 143.99, 141.18, 141.17, 138.27, 125.49, 123.18, 123.10, 122.60, 122.31, 120.91, 70.71, 51.03, 48.28, 48.25, 46.94, 46.83, 35.64, 34.61, 34.55, 31.62, 31.60.

### 2,6-di-tert-butyl-9,10-dihydro-9,10- [3, 4] thiophenoanthracene (6)

To a solution of **5** (1.0 g, 2.65 mmol) in 100 ml of degassed toluene at 120 °C was added dropwise a solution of 2,3-dichloro-4,5-dicyano-benzoquinone (1.33 g, 5.84 mmol) in 50 ml of toluene. After complete addition, the reaction mixture was stirred at 120 °C for 1 h. The reaction mixture was then cooled and filtered through silica gel. The solvent was removed under reduced pressure, and the crude product was further purified by silica gel column chromatography using 5% dichloromethane in hexanes. (590 mg of white solid, 60% yield).

$^1H$ NMR (500 MHz, CDCl$_3$) $\delta$ 7.41 (m, 2H), 7.33–7.27 (m, 2H), 7.03 (m, 2H), 6.90 (s, 2H), 5.31 (s, 2H), 1.30 (s, 18H).

$^{13}C$ NMR (126 MHz, CDCl$_3$) $\delta$ 148.38, 147.65, 145.10, 142.36, 123.12, 122.01, 121.08, 114.03, 49.99, 34.62, 31.55, 21.12, 14.25.

### 12,14-dibromo-2,6-di-tert-butyl-9,10-dihydro-9,10- [3,4] thiophenoanthracene (7)

In total, 1.65 g (4.43 mmol) of **6** was dissolved in 45 ml of THF and cooled to 0 °C. In total, 1.74 g (9.74 mmol) of N-Bromosuccinimide was added in portions to the above solution at 0 °C. After complete addition, the solution was warmed to room temperature overnight. The solvent was removed under vacuum to give the crude product which was purified by silica gel chromatography using hexanes (2.1 g, 90%).

$^1H$ NMR (500 MHz, CDCl$_3$) $\delta$ 7.44 (d, $J = 1.9$ Hz, 2H), 7.34 (d, $J = 7.8$ Hz, 2H), 7.11 (dd, $J = 7.8$, 1.9 Hz, 2H), 5.23 (s, 2H), 1.31 (s, 18H).

$^{13}C$ NMR (126 MHz, CDCl$_3$) $\delta$ 149.14, 147.44, 143.72, 140.92, 123.59, 122.56, 121.47, 100.46, 49.20, 34.69, 31.49.

## Typical polymerisation procedures

**pTA-Ph.** To a microwave vial was added **7** (200 mg, 0.377 mmol), 1,4-bis(4,4,5,5-tetramethyl-1,3,2-dioxaborolan-2-yl)benzene (124.46 mg, 0.377 mmol), tetrakis(triphenylphosphine)palladium (22 mg, 5 mol%) and few drops of Aliquat 336. The vial was sealed with septum and was subjected to vacuum followed by nitrogen purge. The process was repeated 3 times and degassed toluene (2 ml) was added to the vial. The vial was purged with nitrogen for 10 min. Two ml of degassed 2 M Na$_2$CO$_3$ aqueous solution was added and purged with nitrogen for 5 min. The vial was heated at 120 °C for 48 h. The polymers were end capped by reacting with phenylboronic acid (0.5eq) for 4 h and finally with 0.5 ml of bromobenzene for 4 h. After cooling to room temperature, the reaction mixture was poured into methanol (50 ml) with stirring. The precipitate was filtered and washed with deionized water and methanol. The precipitated solids were subjected to Soxhlet extraction sequentially with methanol, acetone, Petroleum spirit 40–60 °C and finally with chloroform. The precipitates were collected by filtration and washed with methanol and dried under vacuum. 110 mg, M$_n$−29.4 KDa, M$_w$−42 KDa, PDI−1.43.

$^1H$ NMR (500 MHz, CDCl$_3$) $\delta$ 7.77 (broad m, 4H), 7.61–7.44 (broad m, 4H), 7.23–7.15 (broad m, 2H), 5.74 (broad s, 2H), 1.38 (broad s, 18H).

**pTA-BT**. To a microwave vial was added **7** (200 mg, 0.377 mmol), 4,7-bis(4,4,5,5-tetramethyl-1,3,2-dioxaborolan-2-yl)benzo[c][1,2,5]thiadiazole (146.4 mg, 0.377 mmol), Tris(dibenzylideneacetone)dipalladium (7 mg, 2 mol%), Tri(o-tolyl)-phosphine (19 mg, 16 mol%) and few drops of Aliquat 336. The vial was sealed with septum and was subjected to vacuum followed by nitrogen purge. The process was repeated three times and degassed toluene (2 ml) was added to the vial. The vial was purged with nitrogen for 10 min. Two ml of degassed 2 M Na$_2$CO$_3$ aqueous solution was added and purged with nitrogen for 5 min. The vial was heated at 120 °C for 48 h. The polymers were end capped by reacting with phenylboronic acid (0.5 eq) for 4 h and finally with 0.5 ml of bromobenzene for 4 h. After cooling to room temperature, the reaction mixture was poured into methanol (50 ml) with stirring. The precipitate was filtered and washed with water and methanol. The precipitated solids were subjected to Soxhlet extraction sequentially with methanol, acetone, Petroleum spirit 40–60 °C and finally with chloroform. The chloroform fraction was concentrated and precipitated in methanol. The precipitates were collected by filtration and washed with methanol and dried under vacuum. 125 mg, M$_n$−37.7 KDa, M$_w$−61.7 KDa, PDI−1.64.

$^1H$ NMR (500 MHz, CDCl$_3$) $\delta$ 8.02 (broad s, 2H), 7.62 (broad s, 2H), 7.53 (broad d, $J = 7.8$ Hz, 2H), 7.21 (broad d, $J = 7.8$ Hz, 2H), 5.78 (broad s, 2H), 1.40 (broad s, 18H).

**pTA.** Polymerisation of pTA using FeCl$_3$ as described by previous literature procedures[73]. A microwave vial was charged with Anhydrous FeCl$_3$ (0.14 g, 1.2 mmol, 2.2 equiv.) and **6** (0.3 g, 0.56 mmol, 1 equiv.) was added via a syringe to the suspension. Anhydrous chloroform (1 ml) was injected. The resulting solution was purged with N$_2$ for 30 min and the reaction was heated to 130 °C for 24 h. The reaction mixture was poured into methanol containing 5% concentration of HCl aq. The precipitate was collected by filtration and washed with methanol then Soxhlet extracted with methanol, acetone, hexane and finally chloroform and the solvent was evaporated to give a bright yellow. Yield: 0.18 g (61%). M$_n$−101 K. $^1H$ NMR (400 MHz, CDCl$_3$) $\delta$ 7.53 (s, 3H), 5.84 (s, 1H), 1.35 (d, $J = 6.9$ Hz, 1H), 1.29 (s, 9H), 1.18 (s, 3H), 1.24–1.14 (m, 1H).

**pTA-Th.** A mixture of **7** (51.2 mg, 47.89 µmol), 5,5'-bis(trimethyl-stannyl)-2,2'-bithiophene (26.18 mg, 53.23 µmol), Pd$_2$(dba)$_3$ (0.97 mg, 1.06 µmol) and P(o-tol)$_3$ (1.30 mg, 4.26 µmol) in anhydrous, degassed chlorobenzene (1.2 ml) was degassed under nitrogen for 15 min and heated to 130 °C overnight. After this, a solution of 2-(tributylstannyl)

thiophene (0.1 ml) and $Pd_2(dba)_3$ (1.50 mg, 1.64 µmol) in anhydrous, degassed chlorobenzene (0.5 ml) were added to the polymerisation mixture which was subsequently stirred for 1 h at 130 °C. In total, 0.1 ml of a solution of 2-bromothiophene (0.1 ml) in anhydrous, degassed chlorobenzene (0.5 ml) was then subsequently added and stirred at 130 °C for a further 1 h. The polymerisation mixture was allowed to cool to room temperature and then precipitated in ethyl acetate (200 ml), followed by the addition of hexane (100 ml). The blue solid was then collected in a thimble and purified via Soxhlet extraction with methanol, acetone, hexane, THF and chloroform. The polymer dissolved in hot chloroform and was then re-precipitated in ethyl acetate (200 ml), with the addition of hexane (100 ml) to give the title polymer (48.1 mg, 86%) as a dark blue solid. $^1$H NMR (500 MHz, 313 K, chloroform-$d$) $\delta$ 8.83 (s), 7.35 (d), 4.44 (br s), 3.87–3.24 (m), 1.53 (br s). GPC (chloroform, 40 °C): $M_n$−23.9 kDa, $M_w$−48.6 kDa.

## Data availability

The data that support the findings of this study are available from the corresponding author upon request. Source data are provided with this paper.

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

## Acknowledgements

The authors would like to acknowledge financial support from KAUST Office of Sponsored Research CRG10, by EU Horizon 2020 grant agreement no. 952911, BOOSTER, grant agreement no. 862474, RoLA-FLEX, and grant agreement no. 101007084 CITYSOLAR, as well as EPSRC Projects EP/T026219/1, EP/W017091/1 and EP/S030727/1.

The authors would like to acknowledge the Henry Royce Institute (through UK Engineering and Physical Science Research Council grant EP/R010145/1) for capital equipment, For the purpose of Open Access, the author has applied a CC BY public copyright licence to any Author Accepted Manuscript (AAM) version arising from this submission.

## Author contributions

F.M. developed the idea, designed and performed experiments, developed analytical tools, analysed data, designed and built testing set-up and wrote the paper. W.Z., B.P. and M.A. designed and synthesised polymers. S.G.C. designed and performed experiments. C.M.A. imaged samples and assisted with manuscript preparation. B.W., F.C., Y.L., J.K., M.N. and H.C. performed experiments. J.T. performed DFT studies. J.S.G. performed XPS studies. E.R., Y.B. supervised by S.E. performed isotope labelled experiments. L.S., J.R.D. and I.M. supervised the project.

## Competing interests

The authors declare no competing interests.
