## [Peer Review File · Nature Communications]

Solution-Processable Polymers of Intrinsic Microporosity for Gas-Phase Carbon Dioxide PhotoreductionREVIEWER COMMENTS

Reviewer #1 (Remarks to the Author):

The work by Moruzzi et al. describes the synthesis of for different photocatalytic PIMs, which are used for the reduction of CO₂ to CO. Reasonable yields for the photoreduction are described but there are some general problems with the report as listed below.

- The formation of solution-processable photocatalytic PIMs has been recently reported but has not been cited by this work. Although the incorporation of the photocatalytic species is significantly different in this work than previously reported the general concept has been reported.
 - o Liu, R.Y., Guo, S., Luo, SX.L. et al. Solution-processable microporous polymer platform for heterogenization of diverse photoredox catalysts. Nat Commun 13, 2775 (2022).
- The photocatalytic performance of polymers with intrinsic porosity should be compared to the linear equivalents to show that the porosity makes a difference. Previously linear photocatalytic polymers have been produced that are like the reported structures here which have displayed high catalytic activity it is therefore unclear what difference the intrinsic porosity has.
- Can the authors comment on why pTA is so much bigger than the other PIMs and is this a fair comparison?
- There is a problem with the PL data. Why was a fixed wavelength of 404nm selected? From the UV-Vis data it is clear that this is not great for some of the polymers.
- Can the authors comment on why pTA is so blue-shifted compared to pTa-Th?
- Moreover, why is pTA-BT so red-shifted compared to just the small molecule equivalent? Angew.Chem.Int.Ed.2016,55,9783–9787
- The surface area of the PIMs should be calculated using NMR not by drying the material and doing BET as the structure between the dry and solubilized state is likely to be very different.
- It is not appropriate to give the average pore size to the angstrom level, especially with the way the incremental pore volume graphs look.
- Can the authors explain the presence of the large volume pore observed by BET and explain why it is not present in the pTA -BT sample?
- Can the authors explain why a large decrease in daily CO production quantity decreases from day 1 to day 2 in figure 3b
- Although it is of interest to investigate the factors influencing photocatalysis I am finding it hard to draw any conclusions from figure four and it should either be further explained in the text or more likely removed

Reviewer #2 (Remarks to the Author):

The authors report four new conjugated polymers with intrinsic porosity and tested them for gas-phase carbon dioxide photoreduction. The activity in photoreduction is correlated with the polymers' porosity, optical properties, energy levels, and photoluminescence. It is a remarkable result, that all four polymers

form CO without the addition of any transition metal-containing catalyst. Clearly, the Nature of excited state plays a key role in the structure of these polymers, besides their intrinsic gas uptake (porosity). Even with the addition of copper as the catalytic center, the rate increased “only” by a factor of 2.6, which indirectly indicates the polymer’s efficient performance itself.

The authors interpret the interplay of the different factors, such as Pd-content, BET surface area, pore size, LUMO energy level (=excited state reduction potential), polymer weight Mw, and photoluminescence lifetime, very well. The key of this study is Figure 4 and the authors give comprehensive conclusions based on their large and versatile dataset collected for the materials. Especially the ICP-MS analysis of Pd or Fe content is a key experiment, which they mastered well. Finally, these polymers are designed in a very simple but clever manner. It’s nothing else than a porous version of a poly(thiophene). But the synthetic strategy, the short sequence of reactions, and the good yields contribute to the high impact and outstanding novelty of this paper. The paper is very well written and any potential question while reading is immediately answered by a thorough experimental analysis by the authors. I have rarely seen such a well-organized and comprehensive manuscript with a complete dataset in the very first submission. The scholarly level of the figures is high as well.

In summary, I suggest publication of the manuscript in Nature Communications with only very minimal changes, as listed below.

1. Scheme 1 should depict the trypticene-like molecules in a perspective view, and drawn in three dimensions (also in the following experimental part). Number ranges should include “en dashes”.
2. The formatting of the scientific language in the SI should be improved. Italics in physical quantities; superscript of exponents (line 89, page 3, SI), subscript of “n” in Mn, with a correct unit, not being “K”; consistent use of “en dashes” for number ranges, especially in the NMR data.
3. Could the authors include a short discussion on the interactions of CO₂ with the polymer, possible coordination sites/active sites, and include this in their evaluation of CO formation rates? pTA-BT might be expected as a superior active site for the interactions with CO₂?

Reviewer #3 (Remarks to the Author):

Moruzzi et al. synthesize several thiophene/trypticene conjugated polymers and examine their performance in gas-phase carbon dioxide photoreduction using hydrogen oxidation as a co-reaction. The polymers have high BET surface areas and show activity that appears to increase with larger pore sizes and longer PL lifetimes. Addition of CuI also increases activity, and catalytic activity does not seem to trend with residual Pd or Fe in the polymer films.

In general, the available evidence supports, but does not prove, photocatalytic activity from its polymers. There appears to be a significant error in the analysis of the stability experiment, which

prevents the experiment from being used to prove that the system is definitively catalytic (see below). In addition, there are a fair number of other methodological questions that remain, and the text and figures could have been constructed with more care (many run-on sentences, duplicate Table 1, undefined acronyms like PESA, incorrect references to data in tables, indistinguishable colors in graphs, etc.).

1. The mole calculation in the stability studies that is used to support the claim that CO necessarily arises from CO₂ is misleading. On S17, it says that 0.1 mg of pTA-Ph is equivalent to 0.21 μmol of polymer, which actually means 0.21 μmol of monomer equivalents in the polymer, and one needs to divide by the DP to obtain the moles of polymer. More importantly, there are 30 moles of carbon per one mole of monomer. This means that to be definitively catalytic, the recycled system must produce at least $30 \times 0.21 \text{ μmol} = 6.7 \text{ μmol}$ total CO. However, reading from Figure 3b, over 5 cycles about 0.08 μmol CO is produced per cycle, yielding a total of about 0.4 μmol. The authors must address this discrepancy and should design a different experiment (such as using ¹³CO₂) to argue that their system is definitively catalytic.
2. A citation for benchmarking the B3LYP-D3 functional as applied to aryl oligomer energy and geometry would be helpful in establishing the legitimacy of the calculations
3. Atomic coordinates should be provided for the optimized structures, and frequency calculations should be performed to ensure that the oligomer structures are at energy minima.
4. Figure 4 visually and argumentatively works quite well, but the authors should be cautious about terms like “correlate” when statistical tests are not used.
5. The lifetimes are based on a weighted average for bi- or tri-exponential fits. Why is it expected that (potentially) distinct mechanisms for emission at a given wavelength would all contribute proportionally to the catalytic effects of the polymers?
6. The degree to which free volume changes with time due to physical aging and plasticization in these polymers is potentially very significant to their catalytic performance. It would be good to repeat BET experiments, and potentially catalysis experiments as well, with polymers that were not freshly synthesized.

Please see below a comprehensive point-by-point response of all the Reviewers' comments. The changes in the manuscript and SI are shown in green, while the responses on this document are in blue.

Reviewer #1 (Remarks to the Author):

The work by Moruzzi et al. describes the synthesis of for different photocatalytic PIMs, which are used for the reduction of CO₂ to CO. Reasonable yields for the photoreduction are described but there are some general problems with the report as listed below.

- The formation of solution-processable photocatalytic PIMs has been recently reported but has not been cited by this work. Although the incorporation of the photocatalytic species is significantly different in this work than previously reported the general concept has been reported.

- o Liu, R.Y., Guo, S., Luo, SX.L. et al. Solution-processable microporous polymer platform for heterogenization of diverse photoredox catalyts. Nat Commun 13, 2775 (2022).

This has now been reported as reference 39, to recognise that solution-processable porous polymers have recently drawn interest as a potential class of materials useful in redox reactions.

- The photocatalytic performance of polymers with intrinsic porosity should be compared to the linear equivalents to show that the porosity makes a difference. Previously linear photocatalytic polymers have been produced that are like the reported structures here which have displayed high catalytic activity it is therefore unclear what difference the intrinsic porosity has.

We agree with the reviewer and in fact had included the linear counterpart, pDHT in the SI. This polymer shows a much lower rate of CO production than the polymers containing the rigid unit. The synthesis and BET of pDHT are still presented in the revised SI. pDHT had a surface area of 29 m²g⁻¹, which is an order of magnitude lower than the porous polymers, and its rate of CO production was significantly lower than that of the porous polymers as a result.

In order to make this more clear in the text the following sentence was added under table 2:

“pDHT was synthesised as an analogous, non-triptyl thiophene linear polymer, to show that the rigid unit was required for porosity. The synthesis and properties of this polymer are shown in the SI (Figures S1-3), and the polymer was used as a non-porous control through the study. The polymer exhibited a porosity of 29 m²g⁻¹, which is an order of magnitude lower than the porous polymers, and its rate of CO production was significantly lower than that of the porous polymers as a result.”

”

- Can the authors comment on why pTA is so much bigger than the other PIMs and is this a fair comparison?

We believe this increase is due to an increase in the density of rigid unit per polymer repeat unit. In the other polymers each repeat unit is in alternation with a non-rigid co-monomer, however in pTA the rigid unit is fully the repeat unit. We believe this polymer to be a fair inclusion in this series since other aspects such as the geometries and energetics are probed via the co-polymerisation with other co-monomers.

- There is a problem with the PL data. Why was a fixed wavelength of 404nm selected? From the UV-Vis data it is clear that this is not great for some of the polymers.

The *figure 1c* shows the comparison of the PL spectra of studied polymers films corrected by the absorbed photons at the excitation wavelength 404 nm, aimed to demonstrate the PL intensity difference in pTA-Ph polymer at the same conditions. We explored the emission at different excitation wavelength and no variation in the PL spectra was observed for the polymers pTA-Th and pTA-BT, figure below. Therefore 404 nm wavelength was selected to compare with the time-resolved PL studies using Time-correlated single-photon counting (TCSPC), which has limited LED excitation wavelength: 365 nm, 404 nm, 470 nm and 635 nm in our system.

To complement our PL studies, the PL spectra of pTA-Th and pTA-BT hat another excitation wavelength have been added to the SI in figure S8.

- Can the authors comment on why pTA is so blue-shifted compared to pTa-Th?

This arises from a larger dihedral angle of the homopolymer, imposed by the rigid repeat unit to avoid steric clashing. The pTa-Th does not have any head-head steric twisting, and therefore has a more planar backbone and subsequently red shifted in comparison to the homopolymer. DFT modelling suggests that the homopolymer, PTA, exhibits a coil-like tertiary structure, whereas pTA-Th is more planar..

- Moreover, why is pTA-BT so red-shifted compared to just the small molecule equivalent?
Angew.Chem.Int.Ed.2016,55,9783–9787

The reviewer raises an interesting point. The paper cited by the reviewer was tested using CV in solution rather than PESA. It has been shown that values can vary depending on testing solid vs solution.

The materials are also not chemically equivalent, since the pTA-BT polymer has a 1:1 repeat unit for our backbone, while the small molecule in the paper has a 2:1 ratio of the monomers. This would mean a different push-pull molecular orbital hybridisation and therefore different bandgaps compared to the chemically inequivalent molecule presented in the published paper.

- The surface area of the PIMs should be calculated using NMR not by drying the material and doing BET as the structure between the dry and solubilized state is likely to be very different.

The BET measurements are taken in the solid state, as are the photocatalytic tests, they are not in solution. Therefore, it is likely the species are very similar if not the same. Many recent papers published in this field rely on BET measurements for surface area, especially in microporosity, including the paper mentioned by the reviewer above. (Richard Y. Liu, R.Y., Guo, S., Luo, SX.L. et al. . Nat Commun 13, 2775 (2022)). It is the most widely accepted technique in the field, and we believe to be appropriate for this study.

Other examples include:

- <https://doi.org/10.1002/anie.201905488>
- <https://pubs.acs.org/doi/10.1021/ja8010176>
- <https://www.sciencedirect.com/science/article/abs/pii/S2211339821000976?via%3Dihub>

- <https://pubs.rsc.org/en/content/articlehtml/2021/nr/d1nr05911d>
- <https://onlinelibrary.wiley.com/doi/full/10.1002/aenm.202001935#aenm202001935-bib-0180>
- <https://onlinelibrary.wiley.com/doi/10.1002/anie.201812790>
- <https://pubs.rsc.org/en/content/articlelanding/2019/GC/C9GC03131F>

• It is not appropriate to give the average pore size to the angstrom level, especially with the way the incremental pore volume graphs look.

We have amended in the text and in table 2 the polymers are now reported to the nm level.

• Can the authors explain the presence of the large volume pore observed by BET and explain why it is not present in the pTA -BT sample?

To aid with understanding this question, all pores were set to the same scale to show consistent x-axis.

A possible explanation for this is that the incorporation of BT repeat unit can lead to unique dipole-dipole intermolecular interactions (see J. Am. Chem. Soc. 2011, 133, 8, 2605–2612). This can cause short contacts and significantly different aggregation to other conjugated polymers. This can perhaps be the cause for the lack of larger pores observed.

<https://doi.org/10.1021/acs.accounts.8b00025>

It would be interesting to further investigate this in the future, but we believe further investigation into the morphology is beyond the scope of this study.

• Can the authors explain why a large decrease in daily CO production quantity decreases from day 1 to day 2 in figure 3b

Although the gasses are exchanged and the sample refilled, it is very possible that factors such as: pore clogging, the presence of formed water, saturation of the active sites and degradation of the polymer over time under light illumination in such a reducing atmosphere are likely occurring, leading to a decrease in rate over time. We believe this trend is consistent with photocatalysis behaviour also observed from other groups.

Following papers also observe a similar decrease quite quickly after the first repeat:

<https://doi.org/10.1021/acsami.2c01646> (photocatalytic coupling)

<https://doi.org/10.1039/C8RA07583B> (hydrogen evolution inorganic material)

<https://pubs.acs.org/doi/full/10.1021/acscatal.7b04323> (Z-scheme with CNs)

<https://doi.org/10.1016/j.apcatb.2019.118067> (polymers for H₂ on CN support)

• Although it is of interest to investigate the factors influencing photocatalysis I am finding it had to draw any conclusions from figure four and it should either be further explained in the text or more likely removed

Thank you to the reviewer for this helpful feedback, in order to increase the clarity of the figure we have added more explanations in the text as suggested. We prefer to keep the figure, as we believe Reviewer 3 has found this a beneficial figure, but with clarity and changes recommended by Reviewer 1.

Reviewer #2 (Remarks to the Author):

The authors report four new conjugated polymers with intrinsic porosity and tested them for gas-phase carbon dioxide photoreduction. The activity in photoreduction is correlated with the polymers' porosity, optical properties, energy levels, and photoluminescence. It is a remarkable result, that all four polymers form CO without the addition of any transition metal-containing catalyst. Clearly, the Nature of excited state plays a key role in the structure of these polymers, besides their intrinsic gas uptake (porosity). Even with the addition of copper as the catalytic center, the rate increased "only" by a factor of 2.6, which indirectly indicates the polymer's efficient performance itself.

The authors interpret the interplay of the different factors, such as Pd-content, BET surface area, pore size, LUMO energy level (=excited state reduction potential), polymer weight Mw, and photoluminescence lifetime, very well. The key of this study is Figure 4 and the authors give comprehensive conclusions based on their large and versatile dataset collected for the materials. Especially the ICP-MS analysis of Pd or Fe content is a key experiment, which they mastered well. Finally, these polymers are designed in a very simple but clever manner. It's nothing else than a porous version of a poly(thiophene). But the synthetic strategy, the short sequence of reactions, and the good yields contribute to the high impact and outstanding novelty of this paper. The paper is very well written and any potential question while reading is immediately answered by a thorough experimental analysis by the authors. I have rarely seen such a well-organized and comprehensive manuscript with a complete dataset in the very first submission. The scholarly level of the figures is high as well.

In summary, I suggest publication of the manuscript in Nature Communications with only very minimal changes, as listed below.

1. Scheme 1 should depict the trypticene-like molecules in a perspective view, and drawn in three dimensions (also in the following experimental part). Number ranges should include "en dashes".

Scheme 1 and the experimental have been changed to match the perspective view style-

2. The formatting of the scientific language in the SI should be improved. Italics in physical quantities; superscript of exponents (line 89, page 3, SI), subscript of "n" in Mn, with a correct unit, not being "K"; consistent use of "en dashes" for number ranges, especially in the NMR data.

The authors have addressed the language in the SI

3. Could the authors include a short discussion on the interactions of CO₂ with the polymer, possible coordination sites/active sites, and include this in their evaluation of CO formation rates? pTA-BT might be expected as a superior active site for the interactions with CO₂?

One of the advantages of organic materials is the ability to incorporate functionalities into the structure with ease. Heteroatoms have indeed been shown to interact with CO₂, in photocatalysis but also in carbon capture, and carbon dioxide transport. This is an important property for CO₂ activation and can possibly lead to more product selectivity if an intermediate is bound more strongly than another and can lower the energy needed to dissociate the molecule. In this series, the

percentage of heteroatoms per repeat unit is small, therefore this study focused mainly on other factors such as porosity, rather than the effect of heteroatoms and a more thorough mechanistic insight. It would be of great interest for a future study to incorporate more heteroatoms to act as CO₂ activating groups, and to also offer more insight into the mechanism of CO₂ photoreduction, which is generally poorly understood.

<https://doi.org/10.1021/ja950416q>

<https://doi.org/10.1016/j.ijggc.2019.102930>

Reviewer #3 (Remarks to the Author):

Moruzzi et al. synthesize several thiophene/ptycene conjugated polymers and examine their performance in gas-phase carbon dioxide photoreduction using hydrogen oxidation as a co-reaction. The polymers have high BET surface areas and show activity that appears to increase with larger pore sizes and longer PL lifetimes. Addition of CuI also increases activity, and catalytic activity does not seem to trend with residual Pd or Fe in the polymer films.

In general, the available evidence supports, but does not prove, photocatalytic activity from its polymers. There appears to be a significant error in the analysis of the stability experiment, which prevents the experiment from being used to prove that the system is definitively catalytic (see below). In addition, there are a fair number of other methodological questions that remain, and the text and figures could have been constructed with more care (many run-on sentences, duplicate Table 1, undefined acronyms like PESA, incorrect references to data in tables, Indistinguishable colors in graphs, etc.).

To improve the clarity of the paper, run-in sentences and references to the tables have been amended. Acronyms have been defined and corrected, such as PESA. We have altered the colour scheme; to make the colours more distinguishable. We have gone through the manuscript again and made many improvements to the language and phrasing of text throughout, as can be seen in the track changes version.

1. The mole calculation in the stability studies that is used to support the claim that CO necessarily arises from CO₂ is misleading. On S17, it says that 0.1 mg of pTA-Ph is equivalent to 0.21 μmol of polymer, which actually means 0.21 μmol of monomer equivalents in the polymer, and one needs to divide by the DP to obtain the moles of polymer. More importantly, there are 30 moles of carbon per one mole of monomer. This means that to be definitively catalytic, the recycled system must produce at least $30 \times 0.21 \text{ μmol} = 6.7 \text{ μmol}$ total CO. However, reading from Figure 3b, over 5 cycles about 0.08 μmol CO is produced per cycle, yielding a total of about 0.4 μmol. The authors must address this discrepancy and should design a different experiment (such as using ¹³CO₂) to argue that their system is definitively catalytic.

We agree that the calculation does not give the optimal answer and is misleading. We have removed the section pertaining to this comment "Stability testing (Figure 3b) was carried out on a 0.1 mg sample of pTA-Ph. By the third recycle, the total number of moles of carbon, in the CO product, produced surpassed the number of moles that were present in the polymer sample, demonstrating that the CO produced did not originate from the polymer. A small decrease in reduction rate was

observed over time, likely due to some degradation of the polymer in the reducing atmosphere and illumination during the measurements, as well as build up of water produced, in the pores of the polymers.” and replaced with , C13 isotope labelling studies as suggested by the reviewer. These are shown in S14 and described in the text, the controls done are also explained in further detail in the main manuscript The control testing that was previously included in the SI, was also further explained in the text:

“In order to verify that the CO measured in the photocatalytic experiments arises from CO₂ reduction, and not polymer decomposition, , control experiments and isotope labelling experiments were carried out. Control experiments are outlined in the appendix (S11). The polymers were tested using the following controls (1) Replacing the CO₂ with N₂ (2) Replacing the H₂ with N₂ (3) using only N₂ (4) with operating conditions gasses but no illumination, Very small amounts of CO were detected in these tests, compared to the overall rate obtained using normal operating conditions, supporting that the photoreduction of CO₂ has arisen from polymer photocatalysis. To further support this, ¹³CO studies were carried out, as described in the SI and shown in Figure S14. ¹³CO production was observed after light irradiation overnight with the addition of ¹³CO₂. ¹³CO was not observed in a ¹²CO₂ run nor in a dark run, further demonstrating the catalytic nature of the polymers. “

Since the studies were not optimised by mass, the units in the paper were changed to m², which is a more convention metric for gas-solid phase reactors in the literature.(<https://link.springer.com/article/10.1007/s11244-005-3827-z>)

2. A citation for benchmarking the B3LYP-D3 functional as applied to aryl oligomer energy and geometry would be helpful in establishing the legitimacy of the calculations

This has now been added to the SI.

3. Atomic coordinates should be provided for the optimized structures, and frequency calculations should be performed to ensure that the oligomer structures are at energy minima.

The number of imaginary vibration mode is 0, which shows it's at energy minima. Atomic coordinates have been included in the SI. (S20)

4. Figure 4 visually and argumentatively works quite well, but the authors should be cautious about terms like “correlate” when statistical tests are not used.

We have removed terms like correlate to emphasise similarities in trend instead, replacing them with “the trend of increasing BET surface area did not match the trend of increasing performance”

5. The lifetimes are based on a weighted average for bi- or tri-exponential fits. Why is it expected that (potentially) distinct mechanisms for emission at a given wavelength would all contribute proportionally to the catalytic effects of the polymers?

The emission lifetime showed a multiexponential decay, fitting only to bi or tri exponential functions of time. Indeed, the contribution of the long-lived population in the more active polymers is low (table S7), indicating that after excitation, the polymers undergo a faster exciton decay. However, after the comparison of different parameters our results suggest that the longer exciton emission lifetime in pTA-Ph might increase the probability that the photogenerated exciton separate before recombination under photocatalytic conditions, contributing to increasing its performance.

6. The degree to which free volume changes with time due to physical aging and plasticization in these polymers is potentially very significant to their catalytic performance. It would be good to repeat BET experiments, and potentially catalysis experiments as well, with polymers that were not freshly synthesized.

We have also considered this possibility. The testing for these measurements were done over the space of one year. Measurements done a year after the first experiments (January 2022 vs January 2023) were still within performance error. This suggests quite good overall stability of the polymers to physical aging. We have added this Figure to the SI as S19.

The first BET shows the curve obtained for pTA-Th in February 2020, the second shown the curve obtained from pTA-Th in March 2022, around the same time as testing occurred, presented in the paper. The curves are quite similar with the first yielding an area of 693 and the second 761, which well within the range of BET values in the study.

General changes to also note:

pTA-Th colour scheme was changed to green for increased visibility.

Figure 2: pore diameter axes changed to be the same for clarity.

Performances reported in m-2 rather than g-1.

REVIEWERS' COMMENTS

Reviewer #1 (Remarks to the Author):

Thank you for addressing the comments I had on the original manuscript. I believe the changes that have been made have improved the manuscript and it should now be accepted.

Reviewer #2 (Remarks to the Author):

The team of McCulloch and co-workers present a thoroughly revised manuscript that has substantially improved in quality. All my questions have been answered satisfactorily. The authors should indeed look into non-covalent and potentially covalent interactions of heteroatoms (of the polymer) with CO₂. Future studies and derivatives of such designed CO₂-binding porous polymers would be exciting. Some motifs might include imidazole (as N-nucleophile) or even some ionic liquid-like motifs).

I have also used the opportunity of the revisions to study the critical questions of the other two reviewers including the replies by the authors. I am usually reserved to comment on the discussion with other reviewers, however, in this case, I would like to add a few aspects:

- The BET surface area can only be estimated accurately by BET gas absorption measurements on dry solids. Working in this field, I have not seen accurate measurements of the surface areas of polymers by NMR methods. I might have missed this. Overall, I find the authors' reply sufficient.
- The isotope labeling experiment was crucial. The authors made the right choice following the reviewers' suggestions. The results are clearly in favor of the manuscript.

In summary, I recommend publication of the manuscript in its current form. It will address a broad community with an up-to-date study of most recent interest. This report could also stimulate many scientists to use their polymers as photocatalysts to convert other substrates than CO₂, for example, more complex organic molecules that require photocatalytic transformations. Congratulations to the authors!

Reviewer #3 (Remarks to the Author):

Overall, Moruzzi et al. have done an excellent job either implementing the suggested changes or defending their original approach as appropriate. In particular, I appreciated the responses in the rebuttal to Reviewer 1's questions about the absorbance and PL properties of the polymers, and the new ¹³C experiments combined with the control experiments in S11 do a much better job of proving catalytic activity than the original recycling experiment did alone. As a very minor addition, I understand that the authors have changed the units of the catalytic performance to be reported per unit area instead of per unit mass to reflect literature standards for gas-solid reactors and do not object to this change, but I would suggest that the authors still provide the mass and area of the samples either in the

SI or as additional columns in Table 4 in the main text.

A minor discrepancy: Table 2 cites a 4 nm average pore size for pTA-BT while the main body text below Figure 3 cites 3 nm instead.

REVIEWERS' COMMENTS

Reviewer #1 (Remarks to the Author):

Thank you for addressing the comments I had on the original manuscript. I believe the changes that have been made have improved the manuscript and it should now be accepted.

Reviewer #2 (Remarks to the Author):

The team of McCulloch and co-workers present a thoroughly revised manuscript that has substantially improved in quality. All my questions have been answered satisfactorily. The authors should indeed look into non-covalent and potentially covalent interactions of heteroatoms (of the polymer) with CO₂. Future studies and derivatives of such designed CO₂-binding porous polymers would be exciting. Some motifs might include imidazole (as N-nucleophile) or even some ionic liquid-like motifs).

I have also used the opportunity of the revisions to study the critical questions of the other two reviewers including the replies by the authors. I am usually reserved to comment on the discussion with other reviewers, however, in this case, I would like to add a few aspects:

- The BET surface area can only be estimated accurately by BET gas absorption measurements on dry solids. Working in this field, I have not seen accurate measurements of the surface areas of polymers by NMR methods. I might have missed this. Overall, I find the authors' reply sufficient.
- The isotope labeling experiment was crucial. The authors made the right choice following the reviewers' suggestions. The results are clearly in favor of the manuscript.

In summary, I recommend publication of the manuscript in its current form. It will address a broad community with an up-to-date study of most recent interest. This report could also stimulate many scientists to use their polymers as photocatalysts to convert other substrates than CO₂, for example, more complex organic molecules that require photocatalytic transformations. Congratulations to the authors!

Reviewer #3 (Remarks to the Author):

Overall, Moruzzi et al. have done an excellent job either implementing the suggested changes or defending their original approach as appropriate. In particular, I appreciated the responses in the rebuttal to Reviewer 1's questions about the absorbance and PL properties of the polymers, and the new ¹³C experiments combined with the control experiments in S11 do a much better job of proving catalytic activity than the original recycling experiment did alone. As a very minor addition, I understand that the authors have changed the units of the catalytic performance to be reported per unit area instead of per unit mass to reflect literature standards for gas-solid reactors and do not object to this change, but I would suggest that the authors still provide the mass and area of the samples either in the SI or as additional columns in Table 4 in the main text.

A minor discrepancy: Table 2 cites a 4 nm average pore size for pTA-BT while the main body text below Figure 3 cites 3 nm instead.

The discrepancy in the average pore size of pTA-BT has been addressed. A table including rates per h per g has been added in the SI for completion, upon the suggestion of reviewer 3.

Also to note, in order to keep with the guidelines of the manuscript, in SI Figure 14, the screenshots were replaced with plots showing the same graphs.